# An Evaluation of Systematic Versus Strategically-Placed Camera Traps for Monitoring Feral Cats in New Zealand

**DOI:** 10.3390/ani9090687

**Published:** 2019-09-16

**Authors:** Margaret Nichols, James Ross, Alistair S. Glen, Adrian M. Paterson

**Affiliations:** 1Department of Pest Management and Conservation, Lincoln University, PO Box 85084, Lincoln 7674, New Zealand; James.Ross@lincoln.ac.nz (J.R.); Adrian.Paterson@lincolnuni.ac.nz (A.M.P.); 2Wildlife Ecology & Management, Manaaki Whenua-Landcare Research, Private Bag 92170, Auckland mail center, Auckland 1142, New Zealand; glena@landcareresearch.co.nz

**Keywords:** camera traps, feral cats, occupancy, abundance, habitat

## Abstract

**Simple Summary:**

Feral cats are detrimental to native biodiversity worldwide. In New Zealand, feral cats are well established across much of the pastoral landscape, including forested areas. Feral cats, like many carnivore species, are elusive in their nature, and often occur at low densities, making them difficult to detect. Camera traps are a useful, non-invasive monitoring device, capable of ‘capturing’ feral cats as they pass by. Although cameras provide a wealth of information about animals within their field of view; there remains much to be learned about optimal camera trap placement within a landscape, if maximizing detection probability is the objective. Here, we report the results of two methods of camera trap deployment within similar sites: (1) systematic deployment on a grid and (2) strategic deployment, predominantly favoring habitats with assumed higher cat activity. Using the Royle–Nichols abundance-induced heterogeneity model (RN), which assumes detection probability and animal abundance are linked, we found that more cats were detected by cameras at forest margins than in mixed scrub or open farmland (but only slightly more than in forest locations). If maximizing cat detections is the aim, we recommend that cameras should be placed at the edges of forests (including forest fragments) whenever feasible.

**Abstract:**

We deploy camera traps to monitor feral cat (*Felis catus*) populations at two pastoral sites in Hawke’s Bay, North Island, New Zealand. At Site 1, cameras are deployed at pre-determined GPS points on a 500-m grid, and at Site 2, cameras are strategically deployed with a bias towards forest and forest margin habitat where possible. A portion of cameras are also deployed in open farmland habitat and mixed scrub. We then use the abundance-induced heterogeneity Royle–Nichols model to estimate mean animal abundance and detection probabilities for cameras in each habitat type. Model selection suggests that only cat abundance varies by habitat type. Mean cat abundance is highest at forest margin cameras for both deployment methods (3 cats [95% CI 1.9–4.5] Site 1, and 1.7 cats [95% CI 1.2–2.4] Site 2) but not substantially higher than in forest habitats (1.7 cats [95% CI 0.8–3.6] Site 1, and 1.5 cats [95% CI 1.1–2.0] Site 2). Model selection shows detection probabilities do not vary substantially by habitat (although they are also higher for cameras in forest margins and forest habitats) and are similar between sites (8.6% [95% CI 5.4–13.4] Site 1, and 8.3% [5.8–11.9] Site 2). Cat detections by camera traps are higher when placed in forests and forest margins; thus, strategic placement may be preferable when monitoring feral cats in a pastoral landscape.

## 1. Introduction

Accurate and precise population estimates are necessary to understand the distribution and relative abundance of a target species for wildlife management. Population surveys over large areas are unlikely to detect or count every individual of a population [1,2], especially for cryptic species and those that occur at low densities [3,4]. Often the problem is low rates of detection, i.e., the probability of an individual being detected is much less than 1, and this problem has inspired a variety of different statistical models and sampling designs for estimating population abundance and dynamics over time, such as occupancy modeling [5]. 

Heterogeneity in detection probability at the individual monitoring station level is another issue to account for in estimating abundance [1,6]. Failure to adjust for heterogeneity in detection probabilities assumes uniform abundance throughout the sites, which is also often incorrect [1,7]. Animals will usually be detected more easily where they are more abundant [1]. The detection probability may also vary as a function of season, site heterogeneity (as in habitat complexity), animal behavior, community structure, competitors/predators [8,9], and other environmental factors [3,10]. The abundance-induced heterogeneity Royle–Nichols (RN) model extends the traditional occupancy modelling approach [3] to account for heterogeneity in detection probabilities, stemming from variation in abundance at the sampling unit level [1]. However, in order to obtain accurate and reliable detection probabilities for the cameras at each site, the assumptions of the RN model must be met. Assumptions of single-season, multi-state occupancy models such as the RN include closure within the occupancy state with all potential heterogeneity modelled within the abundance and detection probabilities, independence among sampling units, independence among detection histories, and no species misidentification [11]. 

### 1.1. Camera Trap Deployment 

Carnivores often occur at low densities, have cryptic behavior [9,12], and require sophisticated monitoring and statistical modelling techniques to circumvent issues of low detection [13]. As a result, camera traps have become an increasingly popular tool for providing population estimates for a variety of carnivore species [9,13,14], including feral cats (*Felis catus*) [15,16,17]. However, there is often wide variation in the deployment of camera traps within a landscape (i.e., number of sampling units used at each site and their distance from each other) [18,19]. For example, camera traps may be placed in a variety of ways, such as with a horizontal or vertical orientation [20,21]; baited or unbaited [22,23]; non-biased or biased allocation across a landscape [24], as in systematic grids/transects [9,25,26]; or deliberately placed near likely target species ‘hot spots’, such as trails, roads, and water features [27]. 

### 1.2. Feral Cats as a Target Species 

Feral cats have a deleterious effect on native wildlife, especially in New Zealand, Australia, and many offshore islands [28,29,30]. Thus, they are often targeted as part of routine predator control operations [16,31]. Cats are adaptable and have variable home-ranges that may overlap, depending on resource availability and density [32]. In New Zealand, feral cat home ranges in Hawke’s Bay farmland are estimated at c. 1.9 km^2^ (males) and 0.9 km^2^ (females), with a density of 3–6/km^−2^ [33]. However, in other habitats around New Zealand such as steep forest terrain in the southern North Island, feral cats were found to have linear home ranges of up to 6.34 km (males), 3.83 km (females), and smaller home ranges for females with kittens (0.84–2.0 km) [34]. Cats may prefer a variety of habitats, but most often those that include water sources and a mix of forest cover (both exotic and native) [31,32].

We aim to compare detection probabilities and mean abundance estimates for cameras using different deployment strategies (systematic vs strategic) on two similar pastoral sites. We choose to use an occupancy modelling approach, particularly the RN model, as we assume feral cats utilize some habitat types more than others. However, for the abundance estimates given by the RN model to remain valid, certain assumptions must be met throughout.

## 2. Materials and Methods 

The study took place at two sites on the East Coast of North Island, New Zealand. Site 1, Toronui Station (~39° 0 S, 176° 46′ E; Figure 1a), is a 1600-ha pastoral property with a mix of open farmland and native forest. Site 2 lies within a 26,000-ha portion of the Cape to City ecological restoration area (~39° S, 177° E; Figure 1b). The habitat is similar to that in Site 1, with a mixture of native forest, open farmland, and some semi-urban habitat. Different camera trap models were used at the two sites, as these sites were originally intended for use in separate studies [26].

In June 2014, we placed 40 camera traps (Reconyx PC 900, RECONYX Inc., Holmen, Wisconsin) in Site 1 for 21 days. Cameras were placed systematically on pre-determined grid points, across a 7-km^2^ grid with c. 500-m spacing between individual cameras. Due to the unbiased deployment method used in Site 1, several cameras were deployed in different habitat types, such as forest (both exotic and indigenous), forest margin (any edge between a forest and another habitat), mixed scrub (scrub, rocky areas, or a combination of the above), and open farmland (exposed farmland/paddocks). 

In November 2015, we placed 60 camera traps (Browning Strike Force BTC-5, Prometheus Group, Birmingham, Alabama) in Site 2 for 21 days. Cameras were placed spatially independent of one another (≥ 2-km) according to literature on mean cat home range size [33,35]. Cameras were deployed with a bias towards the forest and forest margin habitat (assumed to be high cat activity areas), however, as well as in mixed scrub and open farmland habitats. Paddocks with large numbers of livestock, including red deer (*Cervus elaphus*), sheep (*Ovis aries*), and cattle (*Bos taurus*), were avoided, to protect the cameras from damage and to reduce the number of non-target images, that we had previously experienced when deploying cameras at Site 1. 

Camera traps at both sites were programmed to capture images in bursts of three with the minimum possible delay between triggers for each camera type (0.5 s for Recoynx, and 5 s for Browning cameras). The minimum time delay between triggers for this model of Browning cameras is 5 s. All images were marked with a date/time stamp. Each camera’s field of view was positioned horizontally, parallel with the ground (10 cm from ground to the base of the camera set on brackets screwed into trees or wooden stakes, and facing south to reduce false triggers from moving light). If necessary, vegetation was cleared from the camera’s field of view, to reduce false triggers from vegetation moving [36]. A perforated vial containing ferret odor (towels impregnated with the scent of a male ferret) was placed 1.5 m in front of all cameras as a scent lure [21,37], and secured with a tent peg to avoid removal by animals. 

### Occupancy Modelling 

We created nightly detection histories for cats per camera trap night (as taken from midnight to midnight) denoted by either a ‘1’ or a ‘0’, respectively. We then implemented the RN model using the detection histories to estimate mean abundance and detection probabilities for each habitat type within each site. We used script adapted from Bengsen (2014) in the package ‘unmarked’ [38] in R version 3.2 [39]. Habitat type was used as a covariate. Each camera location was located in either forest (F), forest margin (FM), mixed scrub (MS), or open farmland (OF). To assess potential variation in detection probability, we used second-order information-theoretic model-selection procedures. The global model allowed both abundance and detection probability estimates to vary according to which habitat types they were deployed in, whereas for the null model these remained constant. The model did not account for variation in camera trap models. 

Site 1 cat detections have been analyzed previously using a spatially-explicit capture recapture model [26,40]. However, the model in the previous study [26] was unable to converge as there was a low number of clustered detections, as this spatial model relies on an animal encountering multiple sampling units with their home range. Thus, a more traditional occupancy model assuming independence of sampling units was deemed more appropriate for the data. 

## 3. Results

At Site 1, 39 cameras remained operative, for a total of 819 trap nights. At Site 2, 57 cameras remained operative for a total of 1197 trap nights (four cameras were removed due to damage from livestock/user error). Camera traps at Site 1 captured a total of 61,416 images, including 2687 non-target wild species (birds, ship rats, mice, hare, possum, hedgehogs, mustelids, pigs, and goats), 36,143 false triggers, and 19,338 livestock. Camera traps at Site 2 captured a total of 87,709 images, including 20,657 of non-target wild species, 56,664 false triggers, and 7271 of livestock. The number of cameras deployed in each habitat type at each site and the average nightly detections of cats can be found in Table 1.

Sites 1 and 2 had similar detection probabilities for feral cats despite variable numbers of camera traps and different deployment strategies (8.6 % [95% CI 5.4–13.4] Site 1, and 8.3 % [5.8–11.9] Site 2). However, the model suggested only cat abundance varied by habitat type, and these results can be seen in Figure 2. Feral cat abundance as seen on camera was highest in forest margin habitats at both sites, and only slightly higher than forest habitat. 

Model output showed little support for the null model with cat abundance varying by habitat type as seen in Table 2. 

## 4. Discussion

As pest management operations extend to larger landscape areas [41], the need for efficient, accurate, and precise monitoring increases. The primary objective of this study was to compare occupancy model estimates using feral cat detections made by camera traps deployed either systematically (Site 1) or strategically (Site 2), in a pastoral landscape. Although different camera trap models were used at the two sites, we do not believe this had a negative impact on our results, as we categorized data into detection/non-detection events per 24-hr period. 

We used the RN occupancy model to examine whether cat abundance varied by habitat type, as well as a full model where abundance and detection probabilities both varied by habitat type. Model selection suggested only cat abundance varied significantly with habitat type, although this was only a little different to both the null model and full model.

The results from the RN model can only be relied upon if all model assumptions are met throughout the study. The literature states that cat home-ranges are highly variable in size and may overlap [32,42,43]. The same camera trap data from Site 1 had also been used in a previous study [26] as a non-treatment control. Previous analysis using a Bayesian approach dependent on spatial correlation of camera trap detections failed to converge due to lower rates of clustered detections than anticipated given the close spacing of ~ 500 m. Although there may be observational spatial independence of detections, abundance estimates at Site 1 are potentially inflated as cats are expected to travel further than 500 m within their home range according to the literature [34]. The RN model relies on the premise that detection probability is mostly driven by abundance at a location. We can assume there is heterogeneity in the abundance of cats throughout different habitat types at both of these sites. Although mean detection probabilities in total were similar for each deployment array, the higher number of cat detections by cameras in forest margins and forest habitats suggests these should be targeted for monitoring in future operations. Additionally, the high number of images of livestock recorded at Site 1, compared to Site 2, resulted in increased footage processing time and higher risk of damage to camera traps. This was mostly avoided at Site 2 with strategic placement away from open farmland that contained livestock. Although the addition of armored security cases for the cameras may have reduced cosmetic damage from livestock, the cameras damaged in this manner were also unusable for our results due to altered placement and skewed field of view from trampling. We list livestock separately from other non-target species, as they were a cause of camera damage at Site 1.

## 5. Conclusions

Estimates for cat abundance at both sites were highest in forest margin and forest habitats for cameras in both deployment arrays. Our study supports the concept that camera traps placed in the ecotones of different habitats may maximize detections of cryptic target species [24]. Other studies also suggest that cats prefer a combination of forest cover [44] and open habitats for hunting [45]. Accordingly, we recommend for future studies and operations, particularly where cats are an apex predator, that cameras be placed strategically in the margins between forest and other habitat types whenever possible to increase detections. 

### Future Research

Further increases in detections will improve accuracy when gauging the success of a control operation. For example, deploying cameras in pairs [15] or in clusters as per Stokeld et al. (2015) could increase detections. While single cameras can be strategically placed in areas that increase their chances of detection, multiple cameras combined with a biased placement may further increase numbers of detections. Further research into different lures such as sound lures, etc. [46], and other social lures may also increase detections of feral cats if they are present. 

## Figures and Tables

**Figure 1 animals-09-00687-f001:**
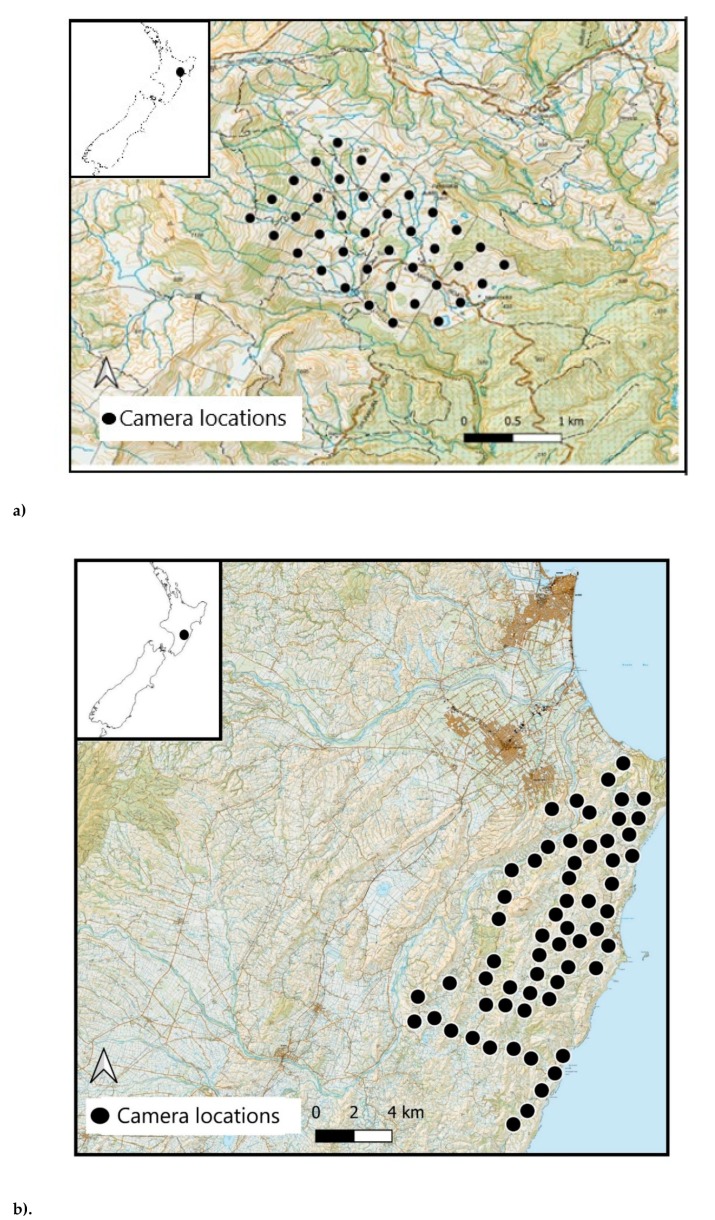
Camera locations at Site 1 (**a**) and Site 2 (**b**), Hawkes bay, North Island, New Zealand.

**Figure 2 animals-09-00687-f002:**
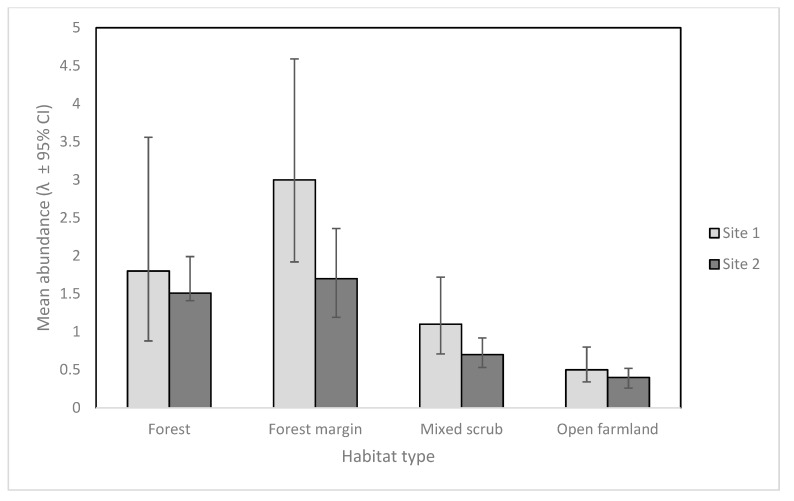
Mean cat abundance (λ) for cameras in each habitat type, (± 95% CI) at Site 1 and Site 2.

**Table 1 animals-09-00687-t001:** Number of cameras and mean nightly cat detections (21 days) per habitat type, for Site 1 and Site 2.

Habitat Type	Number of Cameras (Site 1)	Number of Cameras (Site 2)	Mean Detections by Night (Site 1)	Mean Detections by Night (Site 2)
Forest	2	23	0.2	2.5
Forest Margin	5	12	1	1.5
Mixed scrub	9	15	0.6	0.6
Open farmland	22	7	0.9	0.14

**Table 2 animals-09-00687-t002:** Model selection based on Akaike information criterion that has a correction for small sample sizes (AICc), for Site 1 and Site 2, which includes abundance varying by habitat type (abundance varies), the null model (null), and both abundance and detection probability varying by habitat type (abundance and detection probability vary).

Site 1					
**Model selection based on AICc:**	**AICc**	**Delta AICc**	**AICc Wt**	**Cum.Wt**	**LL**
Abundance varies	361.79	0.0	0.79	0.79	−174.96
Null model	365.19	3.4	0.14	0.93	−180.42
Abundance and detection probability vary	366.69	4.9	0.07	1.00	−172.86
**Site 2**					
**Model selection based on AICc:**	**AICc**	**Delta AICc**	**AICc Wt**	**Cum.Wt**	**LL**
Abundance varies	622.73	0.00	0.64	0.64	−305.78
Null model	624.35	1.62	0.28	0.92	−310.06
Abundance and detection probability vary	626.84	4.11	0.08	1.00	−303.92

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
