# Peer review of "An Evaluation of Systematic Versus Strategically-Placed Camera Traps for Monitoring Feral Cats in New Zealand"

_animals, 2019, doi:10.3390/ani9090687_

Round 1
Reviewer 1 Report
Manuscript title: An evaluation of systematic versus strategically-placed camera traps for monitoring feral cats in New Zealand
Thank you for the opportunity to review this manuscript, which aimed to compare detection probabilities and mean abundance estimates of feral cats when using camera traps using different deployment strategies; specifically, systematic or strategic sampling designs. Understanding where in the landscape to search for target fauna is clearly necessary when research questions or management actions allow for targeted sampling.
That said, there are two key issue that the authors need to address. As the authors mention several times, several assumptions must be met in order to employ the RN model. The important assumption here being independence between sampling units. For site 1, I do not think it has been demonstrated that such an assumption is met. Indeed, the home range size that the authors refer to suggest that 500m spacing is insufficient. Additionally, Fitzgerald and Karl (1986) found linear home range sizes of feral cats to be greater than 3.6 km. Thus, the authors need to better demonstrate that this assumption is met or use a different analytical approach.
The second issue is in the interpretation of models for the second site. The authors concluded that cat abundance is higher in forest margin habitats. The delta AICc values for the two top models are less than two; indicating that although the “abundance varies” model has the most support, the Null model is also reasonably supported. Looking to figure 2, it does seem that the abundance varies between habitats, but most of this variance is between the forest types and the farmland mixed scrub. The standard errors here indicate that there is very weak support for any difference between forest and forest margins.
Generally, the presentation of figure 1 needs to be improved, using R or ArcGIS to present a better map within the study area contexts. Figure 2 should be replotted since it appears at current that the standard error interval for site 2 open farmland is less than 0.
These concerns are certainly solvable with careful consideration.
Fitzgerald, B.M. and Karl, B.J. (1986). Home range of feral cats (Felis catus l.) in forest of the orongorongo valley, Wellington, New Zeland. New Zealand Journal of Ecology, 9, 71–81.
Reviewer 2 Report
In general, a well-written and interesting manuscript. I only have a few comments.
My main concerns, which should be addressed are methodological. How does your model take into account variation in cameras and their capabilities?
Line 94-101
Was there a reason why you used different cameras? It complicated the results as there is a huge difference in delays between cameras. How did you adjust for this? Was it accounted for in the model?
You say - “To assess potential variation in detection probability, we used second-order information-theoretic model-selection procedures.”
Does this procedure take into account trigger speed, variation in the number of cameras, and variation in spacing? If so, say so explicitly.
You also say the following which sounds like the model takes into account only variation in habitat types.
“The global model allowed both abundance and detection probability estimates to vary according to which habitat types they were deployed in, whereas for the null model these remained constant.”
What about independence? Were your images good enough to determine if you saw the same cat on different cameras at site 2? They were placed about the same distance as their homerange.
Are there any future recommendations that could be added to the conclusion? Do you think scent lures helped? Did your technique to contain and secure the lure work? Were there stations where the lure was removed to assess lures vs. no lures?
Did you use security boxes? I use them and have never had cameras damaged but livestock isn’t a big issue at my site.
Any other suggestions to inform future research?
Round 2
Reviewer 1 Report
This is my second review of this manuscript. While the authors have attended to some suggestions, I do not think they have adequately addressed the prior concerns. In fact, in addressing the prior concerns, they have raised new issues.
First, I fail to follow the argument from the authors that the 500 m spacing is sufficient to assume independence between sites. They appear to suggest that while 500 m is insufficient, due to low detections at each site that therefore sites can be considered independent. Perhaps the authors could better explain this line of argument. Failing to find autocorrelation due to a lack of data does not necessarily mean that the sites are independent.
Second, it appears in Figure 2 Mean cat abundance, the error bars have changed considerably from the first iteration of the manuscript. The authors said they “have fixed the error bars to reflect the correct standard errors”. This leads me to ask what was plotted in the first iteration, and how sure can we be that these are in fact the correct error bars and estimates?
Lastly, the maps still need to be improved. A small scale map should be presented to give the reader a general overview of where these study areas are in New Zealand, followed by the larger scale maps of each study site.
Author Response
Firstly, thank you very much for your patient review and improvements to this ms.
Line 198:
We have made a more deliberate effort to explain that the abundance estimates for site 1 may be inflated due to lack of spatial independence in camera trap sites according to average cat home range.
Error bars:
This was a major error on our part, and apologies are required for the confusion. In the previous round of revisions, 95% CI intervals were used instead. Because the 95% CI's were given by the model, and the SE's were calculated by us from the model's output values, we have decided to avoid all subsequent confusion and go back to using the 95% CI's as these were also reported in the abstract results. This has now been amended throughout the document.
Other:
We have updated the maps in Fig. 1a and b to include both up-close and position within the country of NZ.
Again, many thanks for the very helpful improvements throughout!
Round 3
Reviewer 1 Report
Well done to the authors for more or less addressing the issues. The figure of the study area still needs work, I would recommend looking at other examples for guidance and appropriate cartographic representation.
Author Response
Thank you very much for the suggested improvements to the map figures. We hope that the most recent changes are a substantial improvement to the look and layout of theses graphs.
Many thanks